# Protective Effects of Fermented Oyster Extract against RANKL-Induced Osteoclastogenesis through Scavenging ROS Generation in RAW 264.7 Cells

**DOI:** 10.3390/ijms20061439

**Published:** 2019-03-21

**Authors:** Jin-Woo Jeong, Sung Hyun Choi, Min Ho Han, Gi-Young Kim, Cheol Park, Su Hyun Hong, Bae-Jin Lee, Eui Kyun Park, Sung Ok Kim, Sun-Hee Leem, You-Jin Jeon, Yung Hyun Choi

**Affiliations:** 1Freshwater Bioresources Utilization Bureau, Nakdonggang National Institute of Biological Resources, Sangju 37242, Korea; jwjeong@nnibr.re.kr; 2Department of System Management, Korea Lift College, Geochang 50141, Korea; hyunle6869@hanmail.net; 3National Marine Biodiversity Institute of Korea, Seocheon 33662, Korea; mhhan@mabik.re.kr; 4Department of Marine Life Sciences, School of Marine Biomedical Sciences, Jeju National University, Jeju 63243, Korea; immunkim@jejunu.ac.kr (G.-Y.K.); youjinj@jejunu.ac.kr (Y.-J.J.); 5Department of Molecular Biology, College of Natural Sciences, Dong-eui University, Busan 47340, Korea; parkch@deu.ac.kr; 6Department of Biochemistry, Dong-eui University College of Korean Medicine, Busan 47227, Korea; hongsh@deu.ac.kr; 7Anti-Aging Research Center, Dong-eui University, Busan 47227, Korea; 8Marine Bioprocess Co. Ltd., Busan 46048, Korea; hansola82@hanmail.net; 9Department of Oral Pathology and Regenerative Medicine, School of Dentistry, Kyungpook National University, Daegu 41940, Korea; epark@knu.ac.kr; 10Department of Food Science and Biotechnology, College of Engineering, Kyungsung University, Busan 48434, Korea; theresa10000@naver.com; 11Department of Biological Sciences, College of Natural Science, Dong-A University, Busan 49315, Korea; shleem@dau.ac.kr

**Keywords:** fermented oyster, osteoclast differentiation, RANKL, NF-κB, ROS, NOX1

## Abstract

Excessive bone resorption by osteoclasts causes bone loss-related diseases and reactive oxygen species (ROS) act as second messengers in intercellular signaling pathways during osteoclast differentiation. In this study, we explored the protective effects of fermented oyster extract (FO) against receptor activator of nuclear factor-κB (NF-κB) ligand (RANKL)-induced osteoclast differentiation in murine monocyte/macrophage RAW 264.7 cells. Our results showed that FO markedly inhibited RANKL-induced activation of tartrate-resistant acid phosphatase and formation of F-actin ring structure. Mechanistically, FO has been shown to down-regulate RANKL-induced expression of osteoclast-specific markers by blocking the nuclear translocation of NF-κB and the transcriptional activation of nuclear factor of activated T cells c1 (NFATc1) and c-Fos. Furthermore, FO markedly diminished ROS production by RANKL stimulation, which was associated with blocking the expression of nicotinamide adenine dinucleotide phosphate oxidase 1 (NOX1) and its regulatory subunit Rac-1. However, a small interfering RNA (siRNA) targeting *NOX1* suppressed RANKL-induced expression of osteoclast-specific markers and production of ROS and attenuated osteoclast differentiation as in the FO treatment group. Collectively, our findings suggest that FO has anti-osteoclastogenic potential by inactivating the NF-κB-mediated NFATc1 and c-Fos signaling pathways and inhibiting ROS generation, followed by suppression of osteoclast-specific genes. Although further studies are needed to demonstrate efficacy in in vivo animal models, FO may be used as an effective alternative agent for the prevention and treatment of osteoclastogenic bone diseases.

## 1. Introduction

Bone remodeling is a dynamic physiological response coupled with bone formation and resorption, and osteoclasts and osteoblasts have opposite roles in this process. Osteoblasts derived from pluripotent mesenchymal stem cells are involved in bone formation, while osteoclasts are unique differentiated multinucleated giant cells derived from a monocyte/macrophage lineage of hematopoietic stem cells for bone resorbing [1,2]. Therefore, the promotion of excessive bone resorption by activated osteoclasts can lead to various osteolytic lesions and may increase the disability and morbidity of bone disease patients [3,4], indicating that osteoclasts are a primary target for the treatment of diseases associated with excessive bone resorption.

Of the osteoclast differentiation promoting cytokines, the receptor activator of nuclear factor-κB (NF-κB) ligand (RANKL), a member of the tumor necrosis factor (TNF) family, is known to be the most important factor among osteoclast differentiation and activation inducers. The binding of RANKL to the receptor RANK induces activation of the TNF receptor-associated factor 6 (TRAF6) and c-Fos pathways, thereby leading to the activation of various signaling pathways, including NF-κB [5,6]. This promotes the expression of nuclear factor of activated T cell c1 (NFATc1), which in turn stimulates the formation of multinucleated bone-resorbing osteoclasts by enhancing the expression of osteoclast marker genes such as tartrate-resistant acid phosphatase (TRAP) [7,8].

In many previous studies, reactive oxygen species (ROS) have been shown to be an important component as a second messenger for osteoclast differentiation [9,10]. Moreover, recent studies on the inhibition of osteoclast formation and function when RANKL-induced ROS production was blocked by oxidant scavengers suggest that intracellular accumulation of ROS is essential for the differentiation process of osteoclasts [11,12]. In addition, oxidative stress due to excessive production of ROS is known to inhibit osteogenesis by reducing the survival and function of osteoblasts [13,14]. These observations can, at a minimum, explain that preventing excessive ROS production may be a major means of inhibiting osteoclast activity.

There are a variety of drugs currently available in clinical use that can be used to inhibit osteoclast function; however, serious side effects may occur when taken over a long period of time. For example, bisphosphonate, widely used for the treatment of osteoporosis through inhibition of osteoclast function, can cause jaw osteonecrosis and atypical bone fractures and can also increase the risk of heart disease and esophageal cancer [15,16,17]. As another example, denosumab, a humanized monoclonal antibody against RANKL designed to inhibit bone resorption by binding to RANKL, was shown to increase bone mineral density and significantly inhibited the progression of bone erosion in rheumatoid arthritis patients in a randomized controlled trial. However, recent clinical studies have demonstrated that denosumab has also been reported to have various side effects with long-term use that include increased contraction of infectious diseases and arthralgia and muscular pain [15,18,19]. Therefore, in efforts to overcome these side effects, research into the discovery of a substance capable of inhibiting osteoclast differentiation using natural products or other effective extracts derived from natural materials has greatly increased.

It is known that parts of shellfish contain an abundant variety of pharmacologically active substances that may inhibit osteoclast formation and bone loss by promoting osteoblast differentiation [20,21]. Among shellfish, oysters have long been cultivated in many countries and are widely used as a valuable food and industrial resource. In addition, the oyster shell has been used for the purpose of promoting bone health because it contains various minerals [22,23,24]. Recently, extracts or compounds isolated from oysters have been demonstrated to be able to inhibit osteoclast differentiation and induce osteoblast differentiation. For example, in one study, oyster extract and taurine, a major amino acid in oysters, increased growth plate thickness by elevating insulin-like growth factor-1 levels [25]. Recently, Ma et al. [26] reported that a protein named N16, isolated from *Pinctada fucata*, a pearl oyster, inhibited osteoclast differentiation and promoted osteogenic differentiation by inhibiting gene expression associated with mature osteoclasts and inducing osteoblast-specific genes. We recently fermented Pacific oysters (*Crassostrea gigas*) with *Lactobacillus brevis* BJ20 to obtain oyster fermentation extracts with high antioxidant activity. This fermented oyster extract (FO) has been shown to have a protective effect against liver damage that was better than the same effect before fermentation (unpublished data). To test the valuable effects of FO on various human functions, this study was designed to investigate the inhibitory effect of FO on RANKL-induced osteoclast differentiation using murine monocyte/macrophage RAW 264.7 cells as osteoclast precursor cells.

## 2. Results

### 2.1. Cytotoxic Effects of FO in RAW 264.7 Cells

To evaluate the cytotoxicity, we treated RAW 264.7 cells with various concentrations of FO and a 3-(4,5-dimethylthiazol-2-yl)-2,5-diphenyltetrazolium bromide (MTT) assay was performed. As shown in Figure 1A, the cytotoxic effect of FO was not induced at concentrations up to 600 μg/mL, however, significant cytotoxicity was observed in the 800 μg/mL treatment group. Additionally, in the presence of 100 ng/mL RANKL, no significant cytotoxic effect was observed, as compared to untreated control cells at FO concentrations up to 600 ng/mL (Figure 1B). Therefore, the maximum concentration of FO was set at 600 μg/mL in a subsequent study to investigate the effect of FO on RANKL-induced osteoclast differentiation.

### 2.2. FO Suppresses Osteoclast Differentiation in RANKL-stimulated RAW 264.7 Cells

In order to test whether FO inhibits osteoclastogenesis, RAW 264.7 cells were stimulated with 100 ng/mL RANKL in the presence or absence of different concentrations of FO and stained with TRAP. As shown in Figure 2A, osteoclast-like morphological changes in which many cells aggregated and bundled were observed in the RANKL only treatment group. In addition, many multinucleated TRAP-positive cells were produced, and TRAP activity increased in the cells stimulated with RANKL alone (Figure 2B,C), suggesting that RANKL-stimulated RAW 264.7 cells were completely differentiated into osteoclasts. However, FO significantly reduced the number of TRAP-positive multinucleated cells and TRAP activity in a concentration-dependent manner compared to the RANKL-treated group, demonstrating that FO inhibited the fusion and/or differentiation process for osteoclast-like cell formation in RAW 264.7 cells.

### 2.3. FO Disrupts RANKL-induced Formation of F-actin Ring Structure in RAW 264.7 Cells

In order to resorb the mineralized bone surface, which is a major function of mature osteoclasts, the formation of polymerized F-actin rings is essential [27,28]. Therefore, we investigated whether FO inhibited RANKL-induced F-actin ring formation. As indicated in Figure 3, RANKL-treated RAW 264.7 cell-derived mature osteoclasts produced a well-defined F-actin sealing ring with a higher intensity ring height at the cell margin. However, the size of the ring structure in the cells exposed to FO was significantly decreased in a dose-dependent manner compared to the cells treated with RANKL alone, suggesting that FO effectively suppressed osteoclast differentiation and function.

### 2.4. FO Down-Regulates RANKL-Induced Expression of NFATc1, c-Fos, and TRAP mRNA in RAW 264.7 Cells

NFATc1 and c-Fos serve as the most important molecules for inducing transcriptional activation of specific genes, including TRAP, for osteoclast differentiation by RANKL [7,8]. Therefore, in order to evaluate the effect of FO on the transcriptional activity of these genes by RANKL, mRNA expression levels were examined by real-time quantitative polymerase chain reaction (RT-qPCR) analysis. The results showed that the expression of TRAP, as well as NFATc1 and c-Fos, were greatly increased in response to RANKL. However, RANKL-dependent up-regulation of these genes was significantly inhibited in the presence of FO (Figure 4). These results indicate that inhibition of TRAP expression and activity by FO is associated with the inhibition of NFATc1 and c-Fos expression and that they are regulated at the transcription level.

### 2.5. FO Inhibits RANKL-Induced NF-κB Nuclear Translocation and IκBα Degradation in RAW 264.7 Cells

Since the transfer of NF-κB into the nucleus is considered to be a crucial step in osteoclast differentiation, and NF-κB is an important factor affecting the transcription activity of NFATc1 and c-Fos [29,30], we next evaluated whether FO could weaken RANKL-induced activation of NF-κB. As shown in Figure 5A, the expression of NF-κB p65 in the nucleus was remarkably increased within 15 min of RANKL stimulation, whereas the expression of IκBα in cytoplasm was greatly reduced compared to the normal control, indicating that NF-κB was activated by RANKL treatment. However, FO attenuated the RANKL-induced nuclear shift of NF-κB p65 and concomitantly delayed the degradation of IκBα, indicating that FO had an inhibitory effect on the transcriptional activity of NF-κB by RANKL.

### 2.6. FO Suppresses the RANKL-Induced Expression of Osteoclast-Specific Markers in RAW 264.7 Cells

We further examined the effect of FO on the expression of RANKL-induced osteoclast-specific markers such as TRAF6, c-Src tyrosine kinase (c-Src), and phosphatidylinositol 3-kinase (PI3K) for the study of mechanisms involved in the inhibition of osteoclast differentiation by FO. The immunoblotting results showed that upon RANKL treatment, there was a significant increase in the expression of TRAF6, one of the adapter molecules involved in osteoclastogenesis by binding to the RANKL receptor RANK [31] (Figure 5B). However, treatment with FO markedly inhibited RANKL-induced expression of TRAF6. Additionally, we found that, when compared to RANKL treatment, FO treatment dramatically down-regulated the RANKL-induced expression of c-Src, which is required for functional osteoclasts [32]. In addition, when RAW 264.7 cells were treated with RANKL, the phosphorylation of PI3K, which is also required for osteoclast survival and differentiation [33], was dramatically increased. However, FO reduced the phosphorylation level of PI3K induced by RANKL, and no change in PI3K total protein was observed during this process. Moreover, we examined the effects of FO on the levels of the osteoclast-specific markers TRAP and matrix metallopeptidase-9 (MMP-9). Immunoblotting showed RANKL significantly increased levels of these osteoclast-specific markers (Figure 5C), and that FO co-treatment effectively prevented these increases.

### 2.7. FO Alleviates RANKL-Induced Intracellular ROS Production in RAW 264.7 Cells

Because excessive production of intracellular ROS plays an important step for RANKL-induced osteoclastogenesis [9,10], we determined whether FO inhibits RANKL-induced generation of ROS. According to the flow cytometry analysis shown in Figure 6A, the ROS levels in RANKL-stimulated RAW 264.7 cells were markedly increased compared with the control group, and this effect was significantly attenuated in the presence of FO. In addition, N-acetyl cysteine (NAC), a potent ROS scavenger, showed complete inhibition of RANKL-mediated ROS accumulation and osteoclast formation, suggesting that inhibition of osteoclast differentiation by FO may be mediated through a ROS generation blockade.

### 2.8. FO Attenuates RANKL-Induced Expression of Nicotinamide Adenine Dinucleotide Phosphate (NADPH) Oxidase 1 (NOX1) and Rac1 in RAW 264.7 Cells

In the process of osteoclast differentiation by RANKL, the NOX family proteins play a critical role in producing ROS by transferring electrons from NADPH to molecular oxygen [34,35]. In particular, NOX1-dependent ROS generation by RANKL stimulation among NOX subtypes is associated with the activation of TRAF6, c-Src, and PI3K [10,31]. Thus, we examined the effect of FO on the expression of NOX1 and its regulatory subunit, Rac-1 [36,37], in RANKL-stimulated RAW cells. As can be seen from the immunoblot analysis shown in Figure 6B, NOX1 protein expression was dramatically induced by RANKL, which decreased in a concentration-dependent manner in the presence of FO (Figure 6B). Consistent with this result, the increase in Rac-1 expression observed in cells treated with RANKL was greatly weakened by FO.

### 2.9. Inhibition of NOX1 Expression by Small Interference RNA (siRNA) Reduces RANKL-Induced Osteoclastogenesis in RAW 264.7 Cells

In previous studies, ROS is a potent positive regulator for osteolcast differentiation by activating NOX-1 [10,31]. To evaluate whether FO down-regulates ROS production through NOX-1 activation during anti-osteoclastogenesis, we analyzed whether FO decreases ROS production. Therefore, we performed a loss-of-function experiment using siRNA to investigate the role of NOX1 in RANKL-induced ROS generation and osteoclastogenesis. When transfected with NOX1 specific siRNA, as shown in Figure 7A, the expression of NOX1 by RANKL was effectively abolished compared to cells treated with control siRNA. Consequently, similar to the effect of FO, the silencing of NOX1 resulted in a significant down-regulation of RANKL-induced expression of TRAF6 and c-Src and phosphorylation of PI3K (p-PI3K) (Figure 7B), which was related to the elimination of excessive ROS generation (Figure 7C). Moreover, NOX1 siRNA effectively suppressed RANKL-induced osteoclast-like morphological changes and F-actin ring structures, which was also similar to that of FO (Figure 7D), indicating that FO inhibits osteoclastogenesis by suppressing ROS generation, which may be controlled by NOX-1 down-regulation.

## 3. Discussion

In the present study, we investigated the effect of FO on RANKL-mediated osteoclast differentiation using a RAW 264.7 cell model. Our results showed that FO blocked RANKL-induced osteoclastogenesis, as evidenced by the inhibition of TRAP activity and F-actin ring formation associated with inhibition of NF-κB activity, without causing any significant cytotoxicity. In addition, FO reduced the expression of key regulators of osteoclastogenesis (NFATc1 and c-Fos) and the osteoclast-specific genes (TRAF6, c-Src, and PI3K) that were associated with inhibition of NOX1-dependent ROS production.

The reduction of bone density in various bone diseases is caused by the promotion of differentiation and the function of osteoclasts. Therefore, reversing the catastrophic effects caused by enhanced osteoclast formation and maturation is crucial to maintaining normal bone function. RANKL, as a pro-osteoclastogenic cytokine, plays a crucial role in inducing osteoclastogenesis from hematopoietic cells of monocyte-macrophage lineage [29,38]. Terminal differentiation into osteoclasts by RANKL is characterized by the formation of multinucleated giant cells, which is a prior step to form the F-actin loop structure on the bone surface to perform bone resorption [39,40]. Thus, the disruption of RANKL-induced activation and integrity of F-actin ring formation by FO observed in this study strongly suggests that FO can inhibit bone resorption by preventing the early stage of osteoclast differentiation from progenitor cells.

As mentioned in many studies, several cellular signaling pathways for osteoclastogenesis are activated as downstream signals of the RANKL-RANK stimulus. Among them, NF-κB is known to be the most important transcription factor that plays a positive key regulator in the process of RANKL-induced early osteoclast differentiation [29,30]. In its resting state, NF-κB is present in the cytoplasm bound in an inactive form to the IκBα inhibitory protein. However, when IκB-α is phosphorylated by IκB kinase complex and degraded by the ubiquitin-dependent pathway due to the RANKL and RANK binding signal, the released NF-κB translocates from the cytoplasm to the nucleus and triggers transcriptional activation of several osteoclastogenesis-related genes [30,38]. Our immunoblotting results indicated that FO inhibited RANKL-induced nuclear translocation of NF-κB and cytoplasmic degradation of IκB-α, an essential step in NF-κB activation. Although further studies of the relevance of other signaling pathways are needed, the results indicate that blockade of the NF-κB signaling pathway is one of the mechanisms involved in the anti-osteoclastogenic effect of FO against RANKL.

Activation of the NF-κB signaling pathway by the interaction of RANKL and RANK promotes the activation of several downstream transcription factors, such as NFATc1 and c-Fos, a member of the AP-1 transcription factor family [7,8]. According to previous studies, even in the presence of RANKL, NFATc1-deficient embryonic stem cells could not differentiate into osteoclasts, and osteoclast precursor cells could be differentiated into osteoclasts by the exogenous expression of NFATc1 in the absence of RANKL [41,42]. Similar to NFATc1, osteoclast precursor cells lacking c-Fos did not differentiate into mature osteoclasts; however, when overexpressing NFATc1 in c-Fos deficient cells, they differentiated into osteoclasts [8,43]. These observations indicate that NFATc1 acts as a master regulator in the early stage of osteoclast differentiation and in sustained function through enhancing transcription of osteoclast marker genes [5,6]. In the present study, we observed that FO potentially declined RANKL-induced NFATc1 and c-Fos expression. Consistent with the inhibition of NFATc1 and c-Fos, the increased expression of RANKL-induced osteoclast-regulatory and osteoclast-specific genes, including TRAF6, c-Src, p-PI3K, TRAP and MMP-9, were dramatically reduced to the levels of control in the presence of FO. However, it is unclear at present whether FO directly inhibits the expression of NFATc1 and c-Fos through inactivation of NF-κB or whether other signaling pathways that are also required for the activation of NFATc1 are involved. Therefore, although additional experiments are required to determine the precise targets of FO, the current results suggest that inhibition of NFATc1 and c-Fos expression by FO-mediated inactivation of NF-κB may play an important role in decreasing osteoclast-specific gene expression, which in turn might reduce RANKL-induced osteoclast differentiation.

Accumulating evidence indicates that ROS act upstream molecules for promoting the transcription of osteoclast-specific genes at the onset of RANKL-induced osteoclast formation and function [10,13,44]. In contrast, ROS over-generation by oxidative stress prevents osteoblast differentiation and inhibits the proliferation and survival of osteoblasts, resulting in excessive bone loss [13,14], indicating that ROS play a dual role in bone homeostasis. In addition, excessive ROS accumulation due to estrogen deficiency has been shown to be one of the major causes of postmenopausal osteoporosis [45,46]. For this reason, studies on the inhibition of osteoclast differentiation as well as potential targets for osteoporosis treatment have been attracting attention to ROS. According to the results of this study, ROS production by RANKL was distinctly reduced when FO was present. Therefore, we next examined the effects of NAC on RANKL responses, and as expected in the results of previous studies [44,47], when cells were incubated with NAC, RANKL-stimulated ROS production as well as osteoclast differentiation were completely suppressed. From these findings, it can be assumed that FO acts as a scavenger or inhibitor of ROS to prevent osteoclast differentiation.

Over the last decade or more, it has been shown that ROS production by plasma membrane-associated NOX proteins can regulate a variety of biological processes [48,49]. When RANKL binds to its receptor RANK, the NOX family transfers electrons from NADPH to molecular oxygen to form ROS, and this process requires a regulatory protein like Rac1, a small guanosine triphosphatase [36,37]. Although the role of NOX isoforms in osteoclast differentiation is controversial, among the NOX family members, NOX1 is known to participate in osteoclast differentiation in response to RANKL-RANK signaling [9,44,50]. As can be seen from the current results, FO completely blocked the expression of Rac1 as well as NOX1 by RANKL. If NOX1 plays a key role in inhibiting the production of RANKL-mediated ROS by FO, blocking of NOX1 expression should attenuate RANKL-induced osteoclast differentiation. In our results, RANKL-induced ROS production and osteoclast differentiation were markedly suppressed after NOX1-targeted siRNA treatment, similar to the effects of FO, suggesting that the inhibitory potential of FO on RANKL-mediated osteoclastogenesis is associated with the production of NOX1-dependent ROS. Although there is a need for further studies on the involvement of other NOX isotypes and cross-talk with mitochondrial redox signaling, our results indicate that FO inhibits at least RANKL-induced NOX1 expression in osteoclast precursor cells, thereby inhibiting ROS generation and osteoclast differentiation.

In summary, the results of the current study suggest that FO may inhibit RANKL-mediated osteoclast formation through suppressing the NF-κB signaling pathway, NFATc1, and c-Fos expression, and subsequently decreases the expression of osteoclast-specific marker genes. FO further reduced RANKL-induced ROS generation by inhibiting the expression of NOX1. In addition, RANKL-induced ROS production and osteoclast differentiation were significantly attenuated after NOX1-targeted siRNA, suggesting that inhibition of NOX1-dependent ROS generation by FO is involved in the inhibition of RANKL-induced osteoclast differentiation. Although further studies, such as examining the possibility of bone loss prevention using an animal model and the identification of bioactive components of FO, should be performed in the future, the results of this study suggest that FO may have therapeutic potential for bone loss related diseases.

## 4. Materials and Methods

### 4.1. Preparation of FO

The FO used in this study was obtained from Marine Bioprocess Co. Ltd. (Busan, Republic of Korea) and extracted according to a modified version of the method used by Choi et al. [51]. Briefly, the peeled oysters purchased from Deokyeon Seafood Co. Ltd. (Tongyeong, Republic of Korea) were washed with tap water, homogenized, and hydrolyzed at 60 ± 50 °C for 4 h with alcalase (Alcalase^®^ 2.4 L, FG, Brenntag Korea Co., Ltd., Seoul, Republic of Korea). The hydrolyzed oysters were filtered with a vibrating sieve (120 mesh, BUCHI Labortechnik GmbH, Essen, Germany) and concentrated using a rotary evaporator (BUCHI Labortechnik GmbH). For FO manufacturing, *L. brevis* BJ20 (Accession No. KCTC 11377BP) was inoculated into a sterilized seed medium (yeast extract 3%, glucose 1%, monosodium glutamate 1%, water 95%) previously autoclaved at 121 °C for 15 min. Next, 10% (*v*/*v*) of the seed medium cultured at 37 °C for 24 h was inoculated into the culture medium (yeast extract 4%, glucose 1%, monosodium glutamate 6%, hydrolyzed oyster extract 42%, water 47%) and sterilized. After fermentation at 37 °C for 48 h, the mixture was filtered, and the filtrate was concentrated and then spray dried to obtain a powder sample of FO. Prior to use in the experiments, the FO was diluted with cell culture medium to adjust the final treatment concentrations.

### 4.2. Cell Culture and Viability Assay

RAW 264.7 cells were obtained from the American Type Culture Collection (Manassas, VA, USA) and cultured in Dulbecco’s Modified Eagle’s Medium (WelGENE Inc., Daegu, Republic of Korea) containing 10% heat-inactivated fetal bovine serum (WelGENE Inc.), penicillin (100 units/mL), and streptomycin (100 g/mL) at 37 °C in humidified air with 5% CO_2_. For the cell viability study, RAW 264.7 cells were cultured at a density of 1 × 10^4^ cells per well in 96-well plates for 24 h. The cells were treated with RANKL (100 ng/mL; Abcam, Cambridge, MA, USA) in the presence or absence of different concentrations of FO for 48 h. After removing the medium, 0.5 mg/mL MTT (Sigma-Aldrich Chemical Co., St. Louis, MO, USA) solution was added to each well, and the plates were incubated in the dark at 37 °C for 3 h. The supernatant was then replaced with dimethyl sulfoxide (Sigma-Aldrich Chemical Co.) for 10 min to dissolve the blue formazan crystals. The value of optical density was then measured at 450 nm using a microplate reader (Dynatech Laboratories, Chantilly, VA, USA).

### 4.3. Osteoclast Formation and Differentiation Inhibition Assay

To determine the inhibitory effect of FO on osteoclast formation, RAW 264.7 cells were seeded in 48-well plates at a density of 1 × 10^5^ cells per well. After 24 h culture, a medium containing RANKL (100 ng/mL) in the presence or absence of different concentrations of FO was replaced in each well, and the cells were cultured for another five days. The medium containing the relevant reagents was changed every two days during this period. At post incubation, the cultured cells were washed with ice-cold phosphate-buffered saline (PBS) and fixed in 4% paraformaldehyde (pH 7.4; Sigma-Aldrich Chemical Co.) at room temperature for 10 min. The fixed cells were then stained with a commercial TRAP staining kit (Sigma-Aldrich Chemical Co.) according to the manufacturer’s instructions. TRAP-positive multinucleated cells containing more than three nuclei were identified and counted as osteoclasts, and the images were captured using an inverted microscope (Carl Zeiss, Oberkochen, Germany). The experiment was performed in triplicate, and the number of osteoclasts per group was quantified and averaged. In order to measure TRAP activity, the culture medium was collected, and the activity of TRAP was measured using a TRAP assay kit (Sigma-Aldrich Chemical Co.) at 450 nm with an ELISA microplate reader. TRAP activity was calculated as a percentage of the control, in accordance with the method from a previous study [52].

### 4.4. F-Actin Ring Formation Assay

To investigate the inhibitory effect of FO on the formation of F-actin rings in mature osteoclasts, cells cultured under the same conditions as in the differentiation inhibition assay were fixed at room temperature for 10 min by replacing the culture medium with 4% paraformaldehyde solution, as previously described [53]. The cells were washed with ice-cold PBS and then stained with fluorescein isothiocyanate (FITC)-phalloidin solution (Thermo Scientific, Waltham, MA, USA) in darkness for 45 min following a 5 min treatment with a 0.1% Triton X-100 solution to permeabilize the cells. After washing with PBS, the cells were incubated with a 2.5 μg/mL DAPI (Sigma-Aldrich Chemical Co.) solution for 20 min in order to stain the nuclei. The images of F-actin rings were visualized under a fluorescence microscope (Carl Zeiss).

### 4.5. RNA Extraction and RT-qPCR

Total RNA was extracted from cells cultured in the presence or absence of RANKL and FO using the commercially available TRIzol reagent (Sigma-Aldrich Chemical Co.), following the manufacturer’s instructions. After evaluating its quantity and purity using a spectrophotometer, cDNA was synthesized from 1 μg of total RNA using reverse transcriptase with oligo-dT primer (Promega, Madison, WI, USA) per the manufacturer’s recommendations. The cycling parameters used for the amplification of the synthesized cDNA were 40 cycles of denaturation at 95 °C for 15 s and annealing at 60 °C for 60 s. The experiment was performed three times to confirm reproducibility and normalized to the housekeeping gene glyceraldehyde 3-phosphate dehydrogenase (GAPDH).

### 4.6. Protein Extraction and Western Blot Analysis

To extract total cellular proteins, the cells were washed twice with ice-cold PBS and then lysed using the cell lysis buffer before cell debris was removed by centrifugation as previously described [54]. The nuclear and cytosolic proteins were collected using an NE-PER nuclear and cytoplasmic extraction reagents kit (Pierce Biotechnology, Rockford, IL, USA) according to the manufacturer’s instructions. Equal amounts of each protein were subjected to sodium dodecyl sulfate-polyacrylamide gel electrophoresis and transferred to polyvinylidene difluoride membranes (Schleicher and Schuell, Keene, NH, USA). The membranes were blocked with 5% skim milk at room temperature for 1 h and subsequently probed with target primary antibodies with gentle agitation at 4 °C overnight. After washing two times with Tris-buffered saline containing 0.1% Tween-20 for 5 min, the membranes were incubated with the corresponding horseradish peroxidase-conjugated secondary antibodies (Amersham Biosciences, Westborough, MA, USA) for 2 h at room temperature. Protein bands were then visualized using an ECL detection system (Amersham Biosciences).

### 4.7. Determination of Intracellular ROS Levels

The generation of intracellular ROS was monitored through the measurement of the oxidation of DCF-DA to fluorescent DCF by hydroperoxides, as previously described [55]. After the completion of the experimental period, the collected cells were rinsed with PBS and then stained with 10 μM DCF-DA (Sigma-Aldrich Chemical Co.) in darkness at 37 °C for 15 min. Subsequently, the cells were washed twice with PBS and then analyzed using a flow cytometer (BD Biosciences, San Jose, CA, USA) with excitation and emission wavelengths of 485 nm and 530 nm, respectively.

### 4.8. Transfection of siRNA

Small interference RNA (siRNA) targeting NADPH oxidase 1 (NOX1 siRNA) or an equal amount of nonspecific RNA as a control (control siRNA), purchased from Dharmacon, Inc. (Lafayette, CO, USA) were transfected into RAW 264.7 cells using LipofectAMINE 2000 (Invitrogen, Carlsbad, CA, USA) according to the manufacturer’s protocol. Following siRNA transfection for 24 h, RANKL with or without FO was added to the medium and the samples were further cultured for another five days.

### 4.9. Statistical Analysis

All experiments were replicated in three independent experiments. All data were expressed as the mean ± SD and analyzed using the SPSS 13.0 software package (IBM Corp., Armonk, NY, USA). ANOVA with Bonferroni’s multiple comparison test was used to confirm significant differences among the group means. A value of *p* < 0.05 was considered to represent a statistically significant difference.

## Figures and Tables

**Figure 1 ijms-20-01439-f001:**
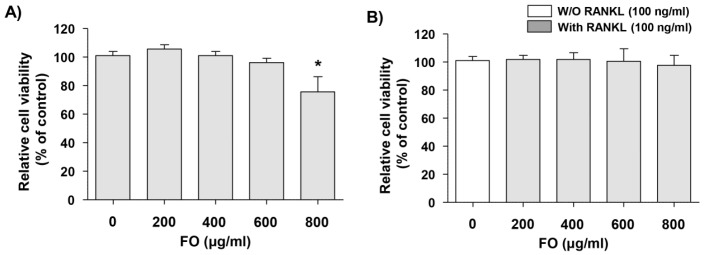
Effects of fermented oyster extract (FO) and RANKL on cell viability in RAW 264.7 cells. Cells were treated with various concentrations of FO in the (**A**) absence or (**B**) presence of 100 ng/mL RANKL for 48 h. Cell viability was assessed by MTT assay. Results are expressed as the percentage of surviving cells over control cells (no addition of FO or RANKL). The results are the mean ± standard deviation (SD) obtained from three independent experiments (* *p* < 0.05 compared with the control group).

**Figure 2 ijms-20-01439-f002:**
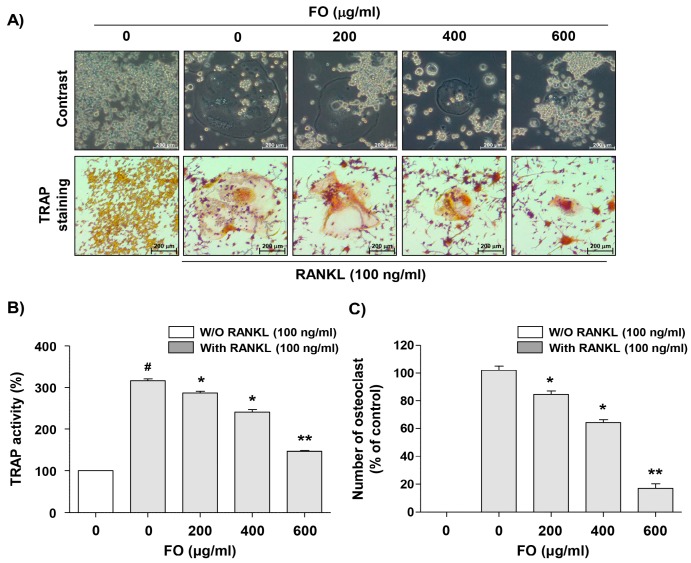
Suppression of RANKL-induced osteoclast differentiation by FO in RAW 264.7 cells. Cells were stimulated with 100 ng/mL RANKL with or without various concentrations of FO for five days. (**A**) The osteoclast cell-like morphology was analyzed, and the cells were fixed and stained for tartrate-resistant acid phosphatase (TRAP) and identified using an inverted microscope. Representative photographs of the morphological changes are presented (Original magnification x100). Scale bar: 200 μm. (**B**) Supernatants were collected from cells grown under the same conditions as (**A**), and the TRAP activity was measured with an ELISA reader. (**C**) TRAP-positive multinucleated cells were counted to determine osteoclast numbers. In (**B**,**C**), each point represents the mean ± SD of three independent experiments (^#^
*p* < 0.05 versus the untreated control; * *p* < 0.05, ** *p* < 0.01 versus the RANKL-treated cells).

**Figure 3 ijms-20-01439-f003:**
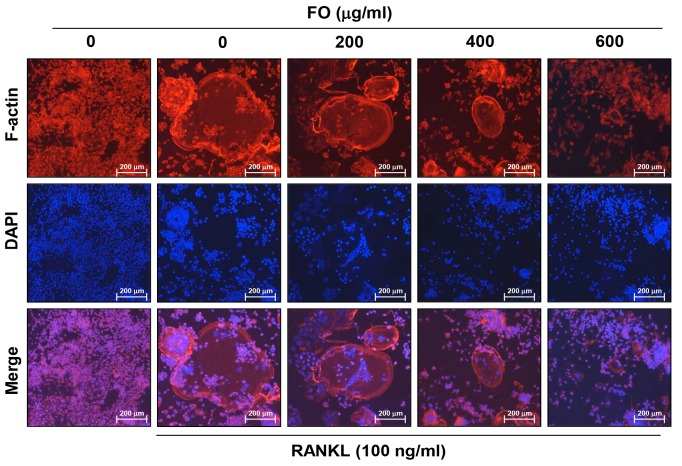
Inhibition of F-actin ring formation by FO in RANKL-stimulated RAW 264.7 cells. The cells were incubated with 100 ng/mL RANKL in the presence or absence of the indicated concentrations of FO for five days. The cells were fixed and stained for F-actin rings with a fluorescein isothiocyanate (FITC)-phalloidin solution. Subsequently, the cells were stained with 4′,6-diamidino-2-phenylindole (DAPI) solution and then imaged with a fluorescence microscope(Original magnification ×100). Scale bar: 200 μm. Representative photographs of the morphological changes are presented.

**Figure 4 ijms-20-01439-f004:**
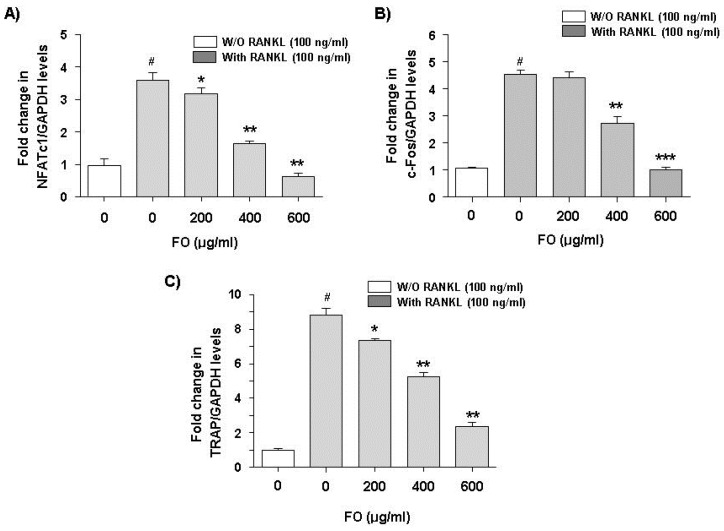
Effects of FO on the expression of NFATc1, c-FOS, and TRAP in RANKL-stimulated RAW 264.7 cells. Cells were treated with the indicated concentrations of FO in the presence or absence of 100 ng/mL RANKL for five days. The relative mRNA expression levels of NFATc1 (**A**), c-FOS (**B**), and TRAP (**C**) were determined by RT-qPCR and normalized to the expression of glyceraldehyde 3-phosphate dehydrogenase (GAPDH). These data are from three separate experiments and are expressed as the mean ± SD (^#^
*p* < 0.05 compared with untreated control group; * *p* < 0.05, ** *p* < 0.01 compared with stimulated group treated with RANKL).

**Figure 5 ijms-20-01439-f005:**
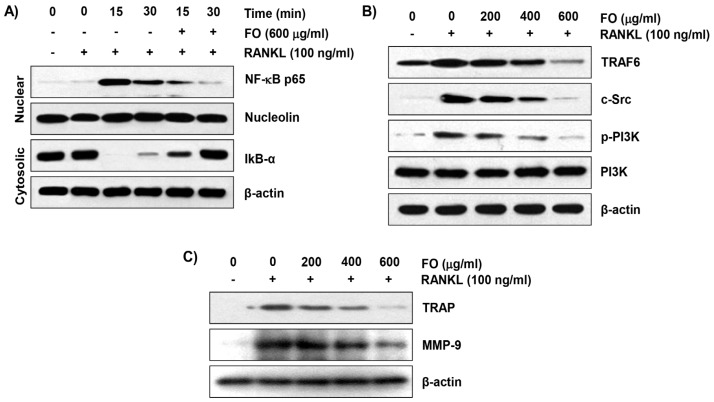
Effects of FO on RANKL-induced activation of nuclear factor-κB (NF-κB) and expression of osteoclast-regulatory genes in RAW 264.7 cells. (**A**) After 100 ng/mL RANKL treatment with or without 600 μg/mL FO for the indicated times, the nuclear and cytosolic proteins were isolated, and the expression of NF-κB and IκBα was determined by Western blot analysis using an enhanced chemiluminescence (ECL) detection system. Nucleolin and β-actin were used as internal controls for the nuclear and cytosolic fractions, respectively. (**B**,**C**) The cellular proteins were isolated from cells cultured under the same conditions, and the expression levels of osteoclast-regulatory and osteoclast-specific proteins were assessed by Western blot analysis; β-actin was used as an internal control. The results shown are representative of three independent experiments.

**Figure 6 ijms-20-01439-f006:**
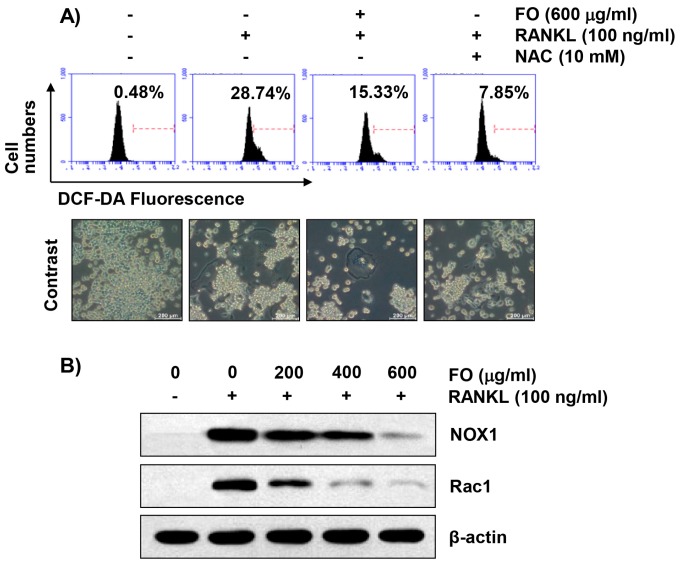
Effects of FO on the RANKL-induced ROS generation and expression of nicotinamide adenine dinucleotide phosphate oxidase 1 (NOX1) and Rac1 in RAW 264.7 cells. The cells were treated with 100 ng/mL RANKL in the presence or absence of 600 μg/mL FO. (**A**) The cells were incubated with 5,6-carboxy-2′,7′-dichlorofluorescein diacetate (DCF-DA), and DCF fluorescence was measured by flow cytometry. The osteoclast cell-like morphology was analyzed using an inverted microscope (Original magnification ×100). Scale bar: 200 μm. (**B**) The cellular proteins were isolated from cells cultured under the same conditions as (**A**), and the expression levels of NOX1 and Rac1 proteins were assessed by Western blot analysis; β-actin was used as an internal control. The results shown are representative of three independent experiments.

**Figure 7 ijms-20-01439-f007:**
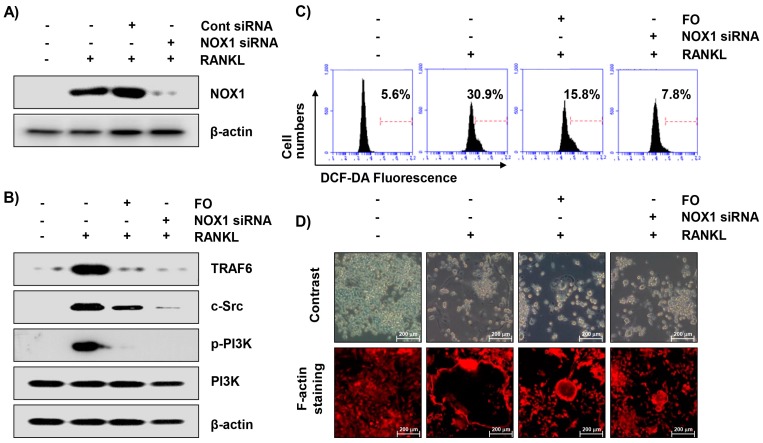
Effects of NOX1 on RANKL-induced ROS generation and osteoclast differentiation in RAW 264.7 cells. RAW 264.7 cells were transiently transfected with a siRNA construct specific for NOX1 or with a control construct, for 24 h. The cells were then treated with 100 ng/mL RANKL in the presence or absence of 600 μg/mL FO for five days. In both (**A**) and (**B**), total cellular proteins were isolated from the cultured cells and immunoblotted with the antibodies indicated. (**C**) The generation of ROS was measured by flow cytometer after 5,6-carboxy-2′,7′-dichlorofluorescein diacetate (DCF-DA) staining. (**D**) The morphological changes of RAW 264.7 cells were observed under an inverted microscope, and the cells were stained with FITC-phalloidin and then imaged with a fluorescence microscope (Original magnification ×100). Scale bar: 200 μm.

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
