# Peer review of "Protective Effects of Fermented Oyster Extract against RANKL-Induced Osteoclastogenesis through Scavenging ROS Generation in RAW 264.7 Cells"

_ijms, 2019, doi:10.3390/ijms20061439_

Round 1
Reviewer 1 Report
In this manuscript, the authors examined effects of fermented oyster extract (FO) on osteoclast formation in vitro. They showed inhibitory effects of FO treatment on osteoclastogenesis along with mRNA induction of NFATc1 and c-fos, nuclear localization of NF-kB, and expression of TRAF6, c-Src and PI3K. They also showed that FO has inhibitory effects on NOX1 induction and ROS production. In addition, NOX1 knockdown exhibited similar effect of FO on osteoclasts. I appreciate the authors efforts to report the novel component. However, I have couple of questions about their interpretation of the results, and I’d like to criticize them.
The authors claimed the effect of FO on in vitro induced osteoclasts that “FO inhibited the fusion process for osteoclast-like cell” (page 3, line 130). I think this is incorrect. Based on the results shown in Figure 2 (B) that FO inhibited TRAP activity itself, Figure 4 that FO inhibited expression of osteoclast master genes, and Figure 5 that FO inhibited induction of critical molecules in osteoclast differentiation, FO seems to block differentiation itself but not fusion process.
The authors explored the effect of FO on ROS production (ROS was sudden showed up and I don’t clearly understand rationale why they focused on ROS here) and showed that NOX1 knockdown was phenocopy of FO treatment (Figure 7), and they concluded that “NOX1-mediated ROS generation is an important mediator in inhibiting RANKL-induced osteoclastogeneis” (page 7, line 247-248). This sentence sounds odd, cause in my understanding the aim of this study was to explore the effect of FO but not NOX1 or ROS. Importance of NOX1 and ROS in osteoclastogenesis is already reported (reference No.44), so the findings (about NOX1 and ROS) are not even novel.
I would assume that the message what the authors want to say is scavenging ROS is the main mechanism of inhibitory effect of FO, since it is the title of this manuscript. If so, can the inhibitory effect of FO be restored by oxidative stress such as H2O2 or AAPH treatment (described in the reference No.9)? Can the induction of TRAF6-c-Src-PI3K, activation of NF-kB, and induction of NFATc1 be rescued by oxidative stress, too?
In Figure 3, the authors examined the effect of FO on actin ring formation but it lacks quantitative results. The effect on actin ring formation should be able to quantify by bone resorbing activity.
In Figure 1, legends described (A) and (B) seems to be opposite.
In Figure 6, the authors showed one (probably representative) flow result of two independent experiments but showed mean values of two experiments. This description is not correct. They should show mean values in a table, or show values of the representative experiment (not mean values) and describe clearly in legends. Same applies to Figure 7 (C).
Regarding FO preparation, it seemed to manufactured with the method described in page 10. How about the quality of FO among batches? Were there any differences of FO quality among batches?
Author Response
Response to Reviewer 1 Comments
In this manuscript, the authors examined effects of fermented oyster extract (FO) on osteoclast formation in vitro. They showed inhibitory effects of FO treatment on osteoclastogenesis along with mRNA induction of NFATc1 and c-fos, nuclear localization of NF-kB, and expression of TRAF6, c-Src and PI3K. They also showed that FO has inhibitory effects on NOX1 induction and ROS production. In addition, NOX1 knockdown exhibited similar effect of FO on osteoclasts. I appreciate the authors efforts to report the novel component. However, I have couple of questions about their interpretation of the results, and I’d like to criticize them.
The authors claimed the effect of FO on in vitro induced osteoclasts that “FO inhibited the fusion and/or differentiation process for osteoclast-like cell” (page 3, line 130). I think this is incorrect. Based on the results shown in Figure 2 (B) that FO inhibited TRAP activity itself, Figure 4 that FO inhibited expression of osteoclast master genes, and Figure 5 that FO inhibited induction of critical molecules in osteoclast differentiation, FO seems to block differentiation itself but not fusion process.
Response: We are appreciated for your suggestions on our manuscript. In our experimental model, we treated with FO on day 2 after treatment with RANKL. If FO blocked the fusion of preosteoclast cells, all cells need to contact each others for fusion but a number of cells maintain small size of cells, indicating that FO prevents cell differentiation. Moreover, some preosteoclast cells made actin rings which are a marker of cell fusion under the condition of FO treatment, which also suggest that FO can influence on the inhibition of cell fusion. Eventually, we believe that FO functions to hamper the fusion and/or the differentiation of preosteoclast cells.
The authors explored the effect of FO on ROS production (ROS was sudden showed up and I don’t clearly understand rationale why they focused on ROS here) and showed that NOX1 knockdown was phenocopy of FO treatment (Figure 7), and they concluded that “NOX1-mediated ROS generation is an important mediator in inhibiting RANKL-induced osteoclastogeneis” (page 7, line 247-248).
Response: Thank you for your comments. As a background to the study of ROS and NOX1, the following sentences were added to page 7.
→ In previous studies, ROS is a potent positive regulator for osteolcast differentiation by activating NOX-1 [10,31]. To evaluate whether FO downregulates ROS production through NOX-1 activation during anti-osteoclastogenesis, we analyzed whether FO decreased ROS production.
This sentence sounds odd, cause in my understanding the aim of this study was to explore the effect of FO but not NOX1 or ROS. Importance of NOX1 and ROS in osteoclastogenesis is already reported (reference No.44), so the findings (about NOX1 and ROS) are not even novel.
Response: This experiment is for the molecular target of FO in anti-osteoclastogenesis. As shown in this study, we showed that FO negatively regulated RANKL-mediated ROS production and NOX-1 expression. Indirectly, NOX-1 knockdown also showed similar result to FO treatment. Therefore, these data indicate that FO inhibits osteoclastogenesis by suppressing ROS generation, which may be controled by NOX-1 downregulation.
I would assume that the message what the authors want to say is scavenging ROS is the main mechanism of inhibitory effect of FO, since it is the title of this manuscript. If so, can the inhibitory effect of FO be restored by oxidative stress such as H2O2 or AAPH treatment (described in the reference No.9)? Can the induction of TRAF6-c-Src-PI3K, activation of NF-kB, and induction of NFATc1 be rescued by oxidative stress, too?
Response: Thank you for your kind discussion and suggestion. Further experiments are required, but in this manuscript, we showed only the antioxidant effect of FO on the actin ring formation, a specific marker of osteoclastogenesis. Please, understand that we can not do enough experiments on your suggestions because the reversion period for our paper is within 10 days.
In Figure 3, the authors examined the effect of FO on actin ring formation but it lacks quantitative results. The effect on actin ring formation should be able to quantify by bone resorbing activity.
Response: We showed the anti-oxidant effect of FO in actin ring fusion, which is a specific marker of osteoclastogenesis. I think that is enough for this experiment. Please note that it will take a long time to prepare for further experiments in the current situation.
In Figure 1, legends described (A) and (B) seems to be opposite.
Response: This was an error during the writing of the paper. We corrected the sentence you pointed out.
In Figure 6, the authors showed one (probably representative) flow result of two independent experiments but showed mean values of two experiments. This description is not correct. They should show mean values in a table, or show values of the representative experiment (not mean values) and describe clearly in legends. Same applies to Figure 7 (C).
Response: We showed the anti-oxidant effect of FO in actin ring fusion, which is a specific marker of osteoclastogenesis. Even if this result is unsatisfactory, please understand that you can not re-experiment.
Regarding FO preparation, it seemed to manufactured with the method described in page 10. How about the quality of FO among batches? Were there any differences of FO quality among batches?
Response: The FO used in this study was provided by Marine Marine Bioprocess Co. Ltd. (Busan, Republic of Korea) as described in Materials and Methods, and it is clear that there is no differences in FO quality between batches because it was produced in the same process.
→ We are appreciated for your valuable suggestions on our manuscript to improve our study. Again, we would like to acknowledge the lack of additional experiments due to time constraints for reversion. Although we did not give enough answers to each comment, pointing out things we have not considered will be a great help in planning similar experiments in the future.

Reviewer 2 Report
The current work provides an insight about the efficacy of FO on the osteoclastogenesis process which has clinical relevance.
However, additional experiments will be needed to support the hypothesis.
1. It will be important to demonstrate the activity of osteoclasts using bone chips to assess the resorptive effect (i.e. pit formation assay) and osteoassay using calcium phosphate coated plates.
2. Effect of FO on osteoclast marker genes such as MMP-9, TRAP, Cathepsin K.
3. Figure 2 is not clear. Osteoclasts are hard to distinguish. Provide relevant images along with quantified data.
4. Since there is a coupling effect between osteoclasts and osteoblasts, what do you think will happen to osteoblast activity? Provide relevant justifications with experimental proof.
Author Response
Response to Reviewer 2 Comments
The current work provides an insight about the efficacy of FO on the osteoclastogenesis process which has clinical relevance.
However, additional experiments will be needed to support the hypothesis.
1. It will be important to demonstrate the activity of osteoclasts using bone chips to assess the resorptive effect (i.e. pit formation assay) and osteoassay using calcium phosphate coated plates.
Response 1: Thank you for your kind suggestion. This experiment focused on whether FO inhibits the differentiation from preosteoclast to osteoclast and we showed the specific marker of it. TRAP is the most important marker with actin ring formation during osteoclastogenesis. We showed those results in Fig. 1. We acknowledge the importance of the experiment you proposed, but please note that additional experimentation is difficult, as the reversion period of this paper is within 10 days.
2. Effect of FO on osteoclast marker genes such as MMP-9, TRAP, Cathepsin K.
Response 2: Although the effect of FO on the expression of Cathepsin K was not investigated, results of TRAP and MMP-9 were added. We have also mentioned these results in this section.
3. Figure 2 is not clear. Osteoclasts are hard to distinguish. Provide relevant images along with quantified data.
Response 3: Currently, we do not have a program to quantify images. Therefore, please acknowledge that we only showed TRAP activity and number of osteoclast.
4. Since there is a coupling effect between osteoclasts and osteoblasts, what do you think will happen to osteoblast activity? Provide relevant justifications with experimental proof.
Response 4: Thank you for your comments. Our next study is for the effect of FO in osteoblast. At the time, we will try to reveal the effect of FO in coculture system between osteoclast and osteoblast.
→ We are appreciated for your comments and suggestions on our manuscript to improve our study. Again, we would like to acknowledge the lack of additional experiments due to time constraints for reversion.

Reviewer 3 Report
Comments to the author:
It is already known that oyster extract contained zinc that has multi-functions for the body. It is stated that in reproductive failure, embryonic defects, and sperm motility in zinc-deficient mice were improved by oyster extract treatment. The other benefits of oyster extract such as improvement of liver function, immunity, and high antioxidant were already known and recently, we can find the oyster extract supplement in the market easily. However, the possible side effects still remain unclear except for those who are allergic to seafood. This paper provides the new benefit of oyster extract as an anti-osteoclastogenesis yet some relevance of oyster extract antioxidant component as anti-osteoclastogenesis has not been explored the reviewer has some specific concerns and criticisms, which are listed below:
1. Oyster extract is one of the sources of glutathione known for protecting against oxidative stress. In this paper, the author needs to recheck what kind of antioxidant component of the FO extract and what is the antioxidant component that mainly against osteoclastogenesis. Please clarify!
2. In page 6 line 200-201, the figure legend is not consistent (figure 5A). Figure 5A is FO and RANKL treatment in a time-dependent manner but the legend says differently.
3. This paper showed that FO extract effectively inhibits mature osteoclast formation (after 5 days of culture), how about the effect of FO extract in the pre-osteoclast stage? Please compare and Clarify!
4. In this paper, FO extract inhibits RANKL-induced NF-kB nuclear translocation and Ikba degradation and alleviates ROS production (NAC) and attenuates NOX1. Is that mean NAC and NOX1 have direct effect to inhibit NF-kB nuclear translocation? The molecular mechanism of FO inhibits NF-kB nuclear translocation via ROS production still unclear, please clarify!
5. In the material and methods, are the time-dependent manner experiment and concentration-dependent manner experiment have the same methods for culturing the cells? Did the author starve the cells before the treatment? Please clarify!
Author Response
Response to Reviewer 3 Comments
It is already known that oyster extract contained zinc that has multi-functions for the body. It is stated that in reproductive failure, embryonic defects, and sperm motility in zinc-deficient mice were improved by oyster extract treatment. The other benefits of oyster extract such as improvement of liver function, immunity, and high antioxidant were already known and recently, we can find the oyster extract supplement in the market easily. However, the possible side effects still remain unclear except for those who are allergic to seafood. This paper provides the new benefit of oyster extract as an anti-osteoclastogenesis yet some relevance of oyster extract antioxidant component as anti-osteoclastogenesis has not been explored the reviewer has some specific concerns and criticisms, which are listed below:
: We are appreciated for your suggestions on our manuscript. Although we did not give enough answers to each comment, pointing out things we have not considered will be a great help in planning similar experiments in the future.
1. Oyster extract is one of the sources of glutathione known for protecting against oxidative stress. In this paper, the author needs to recheck what kind of antioxidant component of the FO extract and what is the antioxidant component that mainly against osteoclastogenesis. Please clarify!
Response 1: We are appreciated for your comments. We are currently analyzing the detailed components of FO. To date, the results have shown that FO contains high levels of GABA (approximately 7%), which may be a major antioxidant, we have thought. And we are analyzing peptied components in FO at the same time. Based on the results of this study, we will perform additional comparative experiments. Please understand that we are unable to provide an accurate answer to your questions.
2. In page 6 line 200-201, the figure legend is not consistent (figure 5A). Figure 5A is FO and RANKL treatment in a time-dependent manner but the legend says differently.
Response 2: This was an error during the writing of the paper and we corrected that.
3. This paper showed that FO extract effectively inhibits mature osteoclast formation (after 5 days of culture), how about the effect of FO extract in the pre-osteoclast stage? Please compare and Clarify!
Response 3: In this study, we confirmed that FO concomitantly inhibited osteoclast differentiation and formation. In our experimental model, we treated with FO on day 2 after treatment with RANKL for 2 days. If FO blocked the fusion of preosteoclast cells, all cells need to contact each others for fusion but a number of cells maintain small size of cells, indicating that FO prevents cell differentiation. Moreover, some preosteoclast cells made actin rings which are a marker of cell fusion under the condition of FO treatment, which also suggest that FO can influence on the inhibition of cell fusion. Eventually, we believe that FO works to hamper the fusion and/or the differentiation of preosteoclast cells.
4. In this paper, FO extract inhibits RANKL-induced NF-kB nuclear translocation and Ikba degradation and alleviates ROS production (NAC) and attenuates NOX1. Is that mean NAC and NOX1 have direct effect to inhibit NF-kB nuclear translocation? The molecular mechanism of FO inhibits NF-kB nuclear translocation via ROS production still unclear, please clarify!
Response 4: Thank you for your comments. Actually, we are not sure if NAC and NOX-1 knockdown directly inhibits the NF-kB nuclear translocation. However, our data showed that FO significantly reduced ROS production accompanied by NOX-1 inhibition, leading to the inhibition of NF-kB translocation.
5. In the material and methods, are the time-dependent manner experiment and concentration-dependent manner experiment have the same methods for culturing the cells? Did the author starve the cells before the treatment? Please clarify!
Response 5: We did not use starvation before treatment with FO, but all experimental conditions were same. To differentiate preosteoclast cells, we treated cells with RANKL for 2 days and then washed cells before additional RANKL for days in the presence of FO. Actually, the experimental design for FO treatment was same.
→ We are appreciated for your valuable comments and suggestions on our manuscript to improve our study. Again, we would like to acknowledge the lack of additional experiments due to time constraints for reversion.

Round 2
Reviewer 1 Report
I understood that 10 days is not enough to perform additional experiments. I think correct description is important for science, though.
Regarding the sentence “NOX1-mediated ROS generation is an important mediator in inhibiting RANKL-induced osteoclastogeneis” (page 8, line 256-257), as the authors mentioned that the experiment is investigate the target of FO, it would be better you change the sentence as you described in the rebuttal; “FO inhibits osteoclastogenesis by suppressing ROS generation, which may be controlled by NOX-1 downregulation”.
Author Response
Response to Reviewer 1 Comments
I understood that 10 days is not enough to perform additional experiments. I think correct description is important for science, though.
Regarding the sentence “NOX1-mediated ROS generation is an important mediator in inhibiting RANKL-induced osteoclastogeneis” (page 8, line 256-257), as the authors mentioned that the experiment is investigate the target of FO, it would be better you change the sentence as you described in the rebuttal; “FO inhibits osteoclastogenesis by suppressing ROS generation, which may be controlled by NOX-1 downregulation”.
Response: According to your comments, the sentence has been revised.
Thank you for encouraging positive evaluation and improving our research.
Reviewer 2 Report
Overall, the manuscript is clear, coherent and unified.
Suitable Quality? Yes
Conclusions Justified? Yes
Clearly Written? Yes
Author Response
Response to Reviewer 2 Comments
Overall, the manuscript is clear, coherent and unified.
Suitable Quality? Yes
Conclusions Justified? Yes
Clearly Written? Yes
Response: Thank you for encouraging us to improve our study.
Reviewer 3 Report
The reviewer understands that the main problem is the limited time of revision (only 10 days). Therefore, the follow up result for the first comments is awaiting.
Author Response
Response to Reviewer 3 Comments
The reviewer understands that the main problem is the limited time of revision (only 10 days).
Therefore, the follow up result for the first comments is awaiting.
Response: Thanks to the positive evaluation and the careful comments on our manuscript.